# Assessment of NASA airborne laser altimetry data using ground-based GPS data near Summit Station, Greenland

Kelly M. Brunt[1,2], Robert L. Hawley[3], Eric R. Lutz[3], Michael Studinger[2], John G. Sonntag[4,5], Michelle A. Hofton[6], Lauren C. Andrews[7,2], and Thomas A. Neumann[2]

[1]Earth System Science Interdisciplinary Center (ESSIC), University of Maryland, College Park, MD, USA
[2]NASA Goddard Space Flight Center, Greenbelt, MD, USA
[3]Department of Earth Sciences, Dartmouth College, Hanover, NH, USA
[4]AECOM Corporation, Wallops Island, VA, USA
[5]Wallops Flight Facility, NASA Goddard Space Flight Center, Wallops Island, VA, USA
[6]Department of Geographical Sciences, University of Maryland, College Park, MD, USA
[7]Universities Space Research Association (USRA), Columbia, MD, USA

Correspondence to: Kelly M. Brunt (kelly.m.brunt@nasa.gov)

**Abstract.** A series of NASA airborne lidars have been used in support of satellite laser altimetry missions. These airborne laser altimeters have been deployed for satellite instrument development, for spaceborne data validation, and to bridge the
data gap between satellite missions. We used data from ground-based Global Positioning System (GPS) surveys of an 11 km- long track near Summit Station, Greenland, to assess the surface- elevation bias and measurement precision of three airborne laser altimeters, including the Airborne Topographic Mapper (ATM), the Land, Vegetation, and Ice Sensor (LVIS), and the Multiple Altimeter Beam Experimental Lidar (MABEL). Ground-based GPS data from the monthly ground-based traverses, which commenced in 2006, allowed for the assessment of 9 airborne lidar surveys associated with ATM and LVIS
between 2007 and 2016. Surface elevation biases for these altimeters, over the flat, ice-sheet interior, are less than 0.12 m, while assessments of measurement precision are 0.09 m or better. Ground-based GPS positions determined both with and without differential post-processing techniques provided internally consistent solutions. Results from the analyses of ground-based and airborne data provide validation strategy guidance for ICESat-2 elevation and elevation-change data products.

## 1 Introduction

A series of National Aeronautics and Space Administration (NASA) airborne and spaceborne altimeters have a mission to produce a continuous time series of ice-sheet surface-elevation change estimates in an effort to determine the long-term contribution of polar ice sheets to sea-level rise. These missions include the Ice, Cloud, and land Elevation Satellite (ICESat; 2003-2009; Schutz et al., 2005), Operation IceBridge (2009-present; Csatho et al., 2014; Koenig et al., 2010), and Ice, Cloud, and land Elevation Satellite 2 (ICESat-2; scheduled to launch in 2018; Abdalati et al., 2010; Markus et al., 2016).

ICESat's Geoscience Laser Altimeter System (GLAS) was a single-beam instrument that recorded the received laser energy as a waveform. GLAS surface elevations were based on reflected 1064 nm wavelength laser light with a 40 Hz pulse-

repetition frequency. GLAS sampled ~70 m diameter footprints every ~170 m along a series of repeated tracks (Schutz et al., 2005). Early assessments of GLAS, based on ground-based Global Positioning System (GPS) surveys of a large and stable salt flat in Bolivia, have shown absolute surface elevation bias of less than 0.02 m and precision of less than 0.03 m under ideal conditions (Fricker et al., 2005). However, estimates of GLAS surface elevation bias and precision from the latter half

of the satellite mission have been closer to 0.06 and 0.15 m, respectively (Kohler et al., 2013), based on data from a 10,000 km ground-based GPS traverse in East Antarctica, which included the interior and the margin of the ice sheet, where surface roughness and slope compromise the accuracy and precision of satellite laser altimetry (Brunt et al., 2010; 2014). ICESat was operated in 'campaign' mode, with two or three 33-day campaigns occurring annually. Surface elevation biases between the ICESat campaigns ('inter-campaign biases') of up to several centimeters have been found in the data (Borsa et al., 2014;

Hofton et al., 2013; Urban et al., 2012; Siegfried et al., 2011) and should be accounted for when determining ice sheet elevation change rates.

ICESat-2 is the follow-on mission to ICESat. ICESat-2 will carry the Advanced Topographic Laser Altimeter System (ATLAS), a 6-beam, photon-counting laser altimeter, which uses short (< 2 ns) 532 nm wavelength pulses, with a 10 kHz repetition rate. ATLAS will have a ~17 m diameter footprint and a ~0.7 m along-track sampling interval (Abdalati et al.,

2010; Markus et al., 2016). ICESat-2 mission requirements include the determination of ice-sheet elevation change rates to an accuracy of less than or equal to 0.004 m a$^{-1}$ (Markus et al., 2016).

While many large-scale ice-sheet-change studies have been based on a satellite-derived time series (e.g., Velicogna et al., 2014; Shepherd et al., 2012; Zwally et al., 2011; Zwally et al., 2005), airborne laser altimetry has played a critical role in: 1) "bridging the data gap" between the satellite missions, with a focus on areas of significant change and interest (Csatho et al.,

2014; Koenig et al., 2010); 2) satellite data validation (Martin et al., 2005; Hofton et al., 2013); and 3) satellite development (McGill et al., 2013; Brunt et al., 2014; 2016).

Operation IceBridge is bridging the data gap between the ICESat and ICESat-2 missions (Koenig et al., 2010). IceBridge mission requirements include: 1) the measurement of surface elevation with a vertical accuracy of 0.5 m; 2) the accurate detection of annual changes of 0.15 m over sampling distances of 500 m in the ice-sheet interior; and 3) the creation of

datasets for cross-calibration and validation of ice-sheet elevations from satellite lidars. Since 2009, IceBridge has annually surveyed both the Greenland and Antarctic ice sheets, as well as sea ice and Arctic glaciers, with a suite of instruments from a variety of airborne platforms, including the Airborne Topographic Mapper (ATM) and the Land, Vegetation, and Ice Sensor (LVIS; previously referred to as Laser Vegetation Imaging Sensor).

Data from airborne laser altimeters also play a critical role in satellite data validation (Martin et al., 2005; Hofton et al.,

2013). In 2001, prior to its association with IceBridge, ATM was deployed over the western United States and the Antarctic Dry Valleys (Martin et al., 2005) to determine ICESat elevation biases of less than 0.02 m. Similarly, LVIS data were collected over the interior of the Antarctic Ice Sheet in 2009 and 2010 as part of IceBridge to determine ICESat inter-campaign surface-elevation biases (Hofton et al., 2013).

Airborne laser altimeters also play a critical role in satellite development. LVIS has served as the airborne emulator for several space-based concepts and missions, including Global Ecosystem Dynamics Investigation Lidar (GEDI). The Multiple Altimeter Beam Experimental Lidar (MABEL) was developed as an airborne ICESat-2 simulator (McGill et al., 2013). MABEL enabled the development of ICESat-2 geophysical algorithms (Kwok et al., 2014) and provided error analysis of the ATLAS measurement strategy (Brunt et al., 2014).

Such data sets demonstrate the utility of airborne laser altimetry for both enhancing and extending the space-based record of elevation measurements as well as for calibration and validation of data from such missions. However, a comparison of these altimeters, including surface measurement biases and precisions, has not been made over the same ground-survey area. In order to constrain the accuracy and utility of these instruments over ice surfaces, intermediary ground-based observations must be used. Here, we present an assessment of the ice-sheet surface elevation bias and surface measurement precision of three NASA airborne laser datasets used in the development and validation of satellite missions (ATM, LVIS, and MABEL) by performing a direct comparison of these datasets with in situ GPS surveys that have been conducted near the center of the Greenland Ice Sheet, at Summit Station, from 2006 to the present.

## 2 Data

### 2.1 Ground-based GPS surveys

Since August 2006, an 11 km ground-based kinematic GPS survey has been conducted monthly near Summit Station, Greenland (Fig. 1). The survey has been part of a larger long-term observation program funded through the National Science Foundation (NSF). The survey route was designed to follow an ICESat reference ground track (#0412); the survey route intersects the ICESat reference ground track often to enable a large number of data 'crossovers', for direct comparison of ground-based and spaceborne elevations. Data from this survey have been used for ICESat surface elevation validation and have provided an assessment of ICESat inter-campaign surface elevation biases (Siegfried et al., 2011).

The monthly Summit Station ground-based GPS survey represents the most temporally long and dense in situ observation of ice-sheet elevation change. The survey is expected to continue through the ICESat-2 mission to provide a nearly 15 year ground-based dataset. The 11 km GPS survey intersected just 6 km of the ICESat reference ground track (Fig. 1). However, the high temporal resolution and long time series of the ground-based GPS data provide a robust means of validating satellite-derived estimates of ice-sheet elevation change. As such, the exact orbit of ICESat-2, and the resultant satellite ground track, was defined in part based on the location of this survey; similar to ICESat, the survey will intersect approximately 6 km of an ICESat-2 reference ground track (Fig. 1).

The kinematic GPS survey is conducted using a dual-frequency Trimble R7 receiver recording at 0.5 or 1 Hz with a Trimble Zephyr antenna (TRM39105); we note that the kinematic surveys have always been conducted using this equipment. Starting in August 2007, the 'roving' antenna was mounted on a static metal post on a sled towed behind a snowmobile at ~5 m s$^{-1}$ (Siegfried et al., 2011; Fig 2). Current survey protocols call for the survey technician to measure the length of the static

antenna post and the depth of the runners of the sled into the snow surface at the beginning and, usually, at the end of each survey (Table 1). These measurements and the appropriate National Geodetic Survey (NGS) antenna model allow for the calculation of the distance from the phase center of the roving antenna to the surface of the snow (Fig. 2). A continuously operating GPS base station has been installed at Summit Station (3 km east of the start of the survey and 6.5 km southeast of the end of the survey; Fig. 1). For the duration of the survey time series, the base station has been a dual-frequency Trimble NetRS receiver recording at 1 Hz with a Trimble Zephyr Geodetic antenna (TRM41249; Siegfried et al., 2011). Periodically, that station is moved and the base station name is altered to reflect this change (e.g., SUMM prior to July 2009, SMM1 between July 2009 and August 2013, and SMM2 from October 2013 to the present, although the station was not renamed until July 2014). Both the base station and the rover logged data solely from the GPS constellation. The Summit base station GPS data are publicly available for download on the University NAVSTAR Consortium (UNAVCO) Data Archive Interface (http://www.unavco.org/data/gps-gnss/data-access-methods/dai2/app/dai2.html).

When it was logistically possible, the timing of the ground-based survey was coordinated with NASA airborne surveys of the region (Table 1). This allowed for the assessment of airborne lidar performance over ice-sheet interiors (e.g., Brunt et al., 2014). When the timing offset between the airborne and GPS surveys is minimized, assessments of lidar performance are made in the absence of environmental factors (e.g., snow, melt, or wind events) that change the surface and potentially compromise the analysis. Six of the airborne campaigns were offset from the ground-based GPS survey by two days or less, however, three of the campaigns were offset by eight days or more, with the maximum offset being 20 days (Table 1).

## 2.2 Airborne Topographic Mapper (ATM)

ATM (Krabill et al., 2002) is one of the two main airborne laser altimetry systems used by NASA's Operation IceBridge. The current ATM configuration generally consists of a dual instrument configuration, with a wide-scan lidar and a narrow-scan lidar integrated simultaneously. The wide-scan lidar has a full scanning angle of 30° and is generally used over the ice sheets; the narrow-scan lidar, which was first integrated with IceBridge in 2012, has a full scanning angle of 5° and is generally used over sea ice but has also been used for high-altitude land-ice flights. Both ATM lidars are conically scanning, full-waveform systems that transmit 532 nm wavelength 6 ns pulses with a 3 or 5 kHz repetition rate.

ATM has been in operation since 1993. Components of ATM, such as the data system and scanner assembly, have been improved over time. The details of the version of the data system (e.g., '4B'), scanner assembly (e.g., 'T2'), and scanning angle (e.g., '30°') that were used for each airborne survey are captured in Table 2.

ATM surveys over Summit Station were generally conducted using the NASA P-3, but ATM has also been integrated with the NASA C-130 (2015 Arctic campaign; Table 1) and the National Oceanic and Atmospheric Administration (NOAA) P-3 (2016 Arctic campaign; Table 1). Surveys were conducted at a nominal aircraft speed of ~100 m s$^{-1}$, and with a nominal altitude of ~450 m above ground level (AGL). At this air speed, altitude, and repetition frequency, the wide-scan ATM lidar generates a 1-m diameter footprint and a scanning swath width of ~250 m and the narrow-scan ATM lidar generates approximately the same footprint with a scanning swath width of ~40 m (Fig. 3). ATM elevation bias and precision, for the

dual instrument configuration, has been assessed based on crossover analysis and comparisons with elevations derived from ground-based GPS surveys of airport departure aprons. ATM elevation bias and precision estimates are 0.07 m and 0.03 m, respectively (Martin et al., 2012).

We obtained the ATM Level-1B Qfit Elevation and Return Strength data (Krabill, 2013) through the National Snow and Ice Data Center (NSIDC) Operation IceBridge Data Portal (http://nsidc.org/icebridge/portal/) for the 6 flights over the Summit Station GPS ground-survey area (Table 1; Fig. 1). The data files include position information of the surface reflection (latitude, longitude, and elevation) that is derived from the combination of data from the laser systems with data from on-board GPS (Javad) and inertial systems (either Applanix POS AV 510 or 610 systems). Positioning information is derived using differential GPS (DGPS) post-processing techniques. DGPS solutions require both a roving GPS receiver and a static base station. ATM position solutions were determined relative to data from a base station that was installed at the departure airport, and was accomplished in a software package developed by the ATM team at NASA Goddard Space Flight Center (GSFC) called GITAR (GPS Inferred Trajectories for Aircraft and Rockets; Martin, 1991). GITAR is optimized for the polar environment and long baselines. It incorporates data from GPS and GLONASS (since 2011) satellites, as well as data from multiple ground stations, for improved satellite geometry, especially at high latitudes.

## 2.3 Land, Vegetation, and Ice Sensor (LVIS)

LVIS (Blair et al., 1999) is the second main airborne laser altimeter used by Operation IceBridge. It is a swath scanning, full-waveform laser altimeter that transmits 1064 nm wavelength 9 ns pulses, with a 500 to 1500 Hz repetition rate (Blair et al., 1999) and using a scan angle that varies between ±6°. LVIS surveys over Summit Station (Table 1) were conducted using both the NASA P-3 (2007) and the DC-8 (2010) at a nominal aircraft speed of ~100 m s$^{-1}$, and an altitude over Summit Station of ~4600 m AGL. At this air speed, altitude, and repetition frequency, LVIS generates a ~10 m diameter footprint and a scanning swath width of ~1000 m (Hofton et al., 2008; Fig. 3). LVIS long-term (e.g., GPS-related) elevation biases, assessed along 2 repeated several hundred-kilometer-long transects over the Greenland Ice Sheet were found to be better than ±0.05 m with precision estimates at multiple crossover locations that are better than 0.07 m (Hofton et al., 2008).

There were two LVIS flights over the Summit Station GPS ground-survey area (Table 1; Fig. 1): one associated with Operation IceBridge (14 Apr 2010) and one as part of a demonstration dataset for future spaceborne concepts (20 Sept 2007). Similar to ATM, we obtained the IceBridge L2 Geolocated Surface Elevation Product, Version 1.1 (Blair and Hofton, 2015), through the NSIDC Operation IceBridge Data Portal (http://nsidc.org/icebridge/portal/). We obtained the Pre-IceBridge LVIS L2 Geolocated Ground Elevation and Return Energy Quartiles, Version 1 (Blair and Hofton, 2011), through the NSIDC (http://nsidc.org/data/blvis2). These files include position information (latitude, longitude, and elevation) of the lowest reflecting surface in the footprint that is obtained from the combination of laser ranges with laser positioning and pointing information (Hofton et al., 2000). Laser positioning and pointing information are derived from an integrated GPS (either Javad, NovAtel, or Ashtech receivers) and inertial system (either Applanix POS AV 510 or 610 systems) and processed using the commercially-available GrafNav (GPS) and POSPac (inertial) software. The 2007 data used DGPS post-

processing techniques relative to a base station at Kangerlussuaq, Greenland. The 2010 data utilized Precise Point Positioning (PPP) techniques, which do not require a base station, but rather rely on more precise satellite orbit and clock information to determine the position of the roving GPS receiver. Position information for the 2007 and 2010 LVIS campaigns incorporated data from the GPS constellation only.

**2.4 Multiple Altimeter Beam Experimental Lidar (MABEL)**

For completeness, we note that a third NASA laser altimeter has flown over the Summit Station GPS ground-survey area. MABEL is a photon-counting laser altimeter that was developed in support of ICESat-2. In April 2012 it surveyed the Greenland Ice Sheet and Arctic sea ice based out of Keflavik, Iceland; data from this campaign, including analysis of data over the Summit Station ground survey, is presented in Brunt et al. (2014). MABEL is distinct from the other two lidars

assessed here in that it has as many as 24 beams profiling in a linear array (as opposed to the swath methods of ATM and LVIS), perpendicular to the direction of flight.

MABEL transmits 532 and 1064 nm wavelength ~1.5 ns pulses with a variable repetition rate (5 to 25 kHz; McGill et al., 2013). MABEL surveys were conducted using the NASA ER-2 at a nominal aircraft speed of ~200 m s$^{-1}$, and with an altitude over Summit Station of ~16,000 m AGL. At this air speed, altitude, and a 5 kHz repetition frequency, MABEL

generates a 2 m diameter footprint every 0.04 m with a swath width of as much as 2000 m. Based on an error analysis, Brunt et al. (2014) estimate MABEL elevation uncertainty for the Summit Station region to be 0.15 m. MABEL surface elevation bias and measurement precision has been assessed based on direct comparisons of MABEL surface elevations with digital elevation models derived from ground-based GPS data collected on airport departure aprons. MABEL surface measurement precision assessments are generally 0.11 to 0.14 m, but have been as high as 0.24 m (Brunt et al., 2014; Brunt et al., 2016;

Magruder and Brunt, 2016). MABEL surface elevation bias is generally on the order of 1 m; while this bias is relatively large, it is within the mission design goals of MABEL (ICESat-2 algorithm development and error analysis), which focus on surface measurement precision. MABEL data files include position information derived from a GPS integrated with a NovAtel HG1700 AG58 inertial system and are available via the NASA ICESat-2 website (http://icesat-2.gsfc.nasa.gov).

Because MABEL is a multibeam profiling (rather than scanning) lidar, there are relatively few intersections between

MABEL beams and the ground-based GPS survey, ultimately resulting in poor quality statistics (Fig. 3), based on small sample size and poor spatial distribution. These limited areas of airborne and ground-based data intersection are highly clustered in the few places where the MABEL profile crossed the GPS survey, and therefore do not represent a spatially diverse assessment of MABEL instrument performance. Consequently, we exclude MABEL from further discussion, as the dataset is fundamentally different than that of the other scanning lidars considered here.

# 3 Methods

## 3.1 Ground-based GPS survey data processing

Ground-based position solutions from three GPS post-processing software packages, using both PPP and DGPS methods, were compared with airborne elevation data. PPP solutions were acquired using Inertial Explorer v.8.60, a commercial GPS post-processing software package developed by NovAtel. One set of ground-based DGPS solutions was acquired using TRACK (Trajectory Calculation with Kalman filter), the kinematic DGPS component of GAMIT, a GPS utility that was partially developed and supported by the Massachusetts Institute of Technology. Kinematic GPS positions from TRACK v.1.28 software (Chen, 1998) were determined by carrier-phase differential processing relative to the Summit GPS base station. A second set of ground-based DGPS solutions was acquired using ATM's GITAR post-processing software (Martin, 1991). For the DGPS results, the positions of the Summit GPS base station were obtained using GIPSY (GNSS-Inferred Positioning System and Orbit Analysis Simulation Software). For the GITAR solutions, the base station positions represent an average of four days of data, centered on the timing of the ground survey. For the TRACK solutions, the base station positions represent an average recorded over the duration of the ground-based survey.

Independent of post-processing method, all of the ground-based GPS solutions are based on final precise orbit and clock information from the Crustal Dynamics Data Information System (CDDIS) hosted at GSFC. Processing using TRACK corrected for errors associated with the ionosphere by incorporating an IGS data product. To mitigate the effect of multipath distortion, all processing methods used a cut-off angle (7.5°, 10°, and 12° for Inertial Explorer, TRACK and GITAR, respectively). Inertial Explorer and TRACK used a Saastamoinen model to correct for tropospheric delay, while GITAR used a gridded reanalysis data product from the National Centers for Environmental Prediction (NCEP). And all processing methods corrected for solid Earth tides based on an Earth Rotation and Reference System Services, or IERS, model.

All of the ground-based GPS data were solved to the phase center of the antenna. TRACK and PPP solutions used the L1 antenna phase center, while GITAR used the LC phase center. The solutions were then referenced to the ellipsoid (WGS84) and datum of the matching airborne data (either ITRF00, ITRF05, or ITRF08, indicated in Tables 1 and 2). The GPS phase-center elevation solutions were then reduced to the snow surface (Fig. 2) using data from the field (Table 1) and the appropriate NGS antenna model phase-center offsets. Specifically, the calculation of the height of the surface of the snow ($h$) is:

$$h = GPS_{PC} - h_{AntPost} - h_{NGSmodel} + h_{RunnerDepth} , \tag{1}$$

where $GPS_{PC}$ is the surveyed position solution to the phase center of the ground-based roving antenna, $h_{AntPost}$ is the height of the antenna post (1.785 or 1.797 m, depending on the survey; Table 1), $h_{NGSmodel}$ is the NGS model distance between the antenna phase center and the antenna base plane (0.056 or 0.061 m for the L1 or LC phase centers, respectively), and $h_{RunnerDepth}$ is the depth of the sled runners in the snow surface (variable, ranging from 0.0125 to 0.02 m; Table 1). We note that the ground-based GPS data were collected at 1 Hz, with the snowmobile operating at ~5 m s$^{-1}$, giving the GPS data an effective 5 m diameter footprint.

### 3.2 Ground-based GPS and airborne lidar elevation comparison strategies

Once the kinematic GPS data were post-processed and reduced to the snow surface, we compared the ground-based GPS surface elevation data directly to the airborne surface elevation data. We used two different approaches: a 'nearest-neighbor' analysis and a 'zone' analysis.

We note that the footprint sizes of the altimeters are different: for the data used in these analyses, ATM has a ~1 m diameter footprint and LVIS has a ~10 m diameter footprint. When comparing the ground-based GPS data with the lidar data, we chose a search radius around each lidar data point that was equal to the size of the given lidar footprint; this was intended to ensure that the ground elevation data were representative of what the lidar was sampling.

In the nearest-neighbor analysis, we determined the closest single ground-based GPS data point for every lidar data point.
Then we limited our analysis to points where the lidar and GPS measurements were within the search radius that was appropriate for the given lidar. We then assessed the difference between the lidar surface elevation and the closest GPS surface elevation for the data that met the search criteria.

In the zone analysis, we identified every ground-based GPS data point within the appropriate search radius around the lidar data coordinates (which represent the center of the lidar footprint); not every lidar data point had GPS data that met this
search criteria. Then we determined the mean of the GPS elevations within this 'zone'. Similar to the nearest-neighbor analysis, we then assessed the difference between the lidar surface elevation data point and the mean of the GPS surface elevations within the zone.

For each airborne mission analysis, once the ground-based GPS surface elevation data ($GPS_{elevation}$) were associated with the lidar surface elevation data ($Lidar_{elevation}$), the mean elevation difference is the lidar elevation bias ($B$):

$$B = \frac{\sum(Lidar_{elevation} - GPS_{elevation})}{N},$$  (2)

where $N$ is the total number of either the nearest-neighbor data points, or the total number of zones, that met the distance criteria. This lidar bias is relative to the ground-based GPS elevation data, which we are taking to be the truth. By assuming that the ground-based GPS data represent truth, for these analyses we assume their errors are zero. In actuality, these errors are not zero and are a function of several terms, including: 1) formal errors, which vary based on processing methods and
include factors such as ephemeris and clock errors; 2) atmospheric errors, associated with both the ionosphere and troposphere; 3) multipath errors; 4) the precision of the base station estimate to which the survey is related (in the case of the DGPS processing methods); and 5) observational errors such as variable penetration of the sled into the snow along the course of the survey. We note that the existing ground-based and airborne elevation data are likely correlated, as they are based on similar GPS measurement and, in the case of GITAR, processing strategies. The standard deviation of the bias ($B$)
in Equation 2 is the spread of the data about the mean, taken to be the lidar surface measurement precision. Surface measurement precision is defined here as the vertical dispersion of the lidar measurements about the mean surface and takes into account properties of the surface that will affect the measurement (e.g., slope and roughness) and altimeter precision,

which is a function of several terms, including: 1) geolocation errors, which are a function of all of the GPS terms described above, inertial measurement errors, altitude, and horizontal uncertainty; 2) errors in altimeter timing; 3) the size of the footprint on the surface, which is a function of altitude and beam divergence; and 4) lidar data processing errors. Over the relatively smooth and flat ice found in the Summit Station region, these surface effects and instrument effects are not easily distinguished from one another in the lidar surface measurement data.

## 4 Results

To assess the ground-based GPS post-processing methods used in this analysis, we compared data from a unique ground-based survey that conducted two separate passes of the traverse route on 5 May 2009 (Table 1). We compared the second pass to the first pass, using a nearest-neighbor approach, and calculated the mean elevation residual for 1067 points. For the DGPS methods, the TRACK residual was 0.004 m (standard deviation 0.055 m), while the GITAR residual was 0.026 m (standard deviation 0.058 m). For the PPP method, this residual was -0.009 m (standard deviation 0.057 m). Thus, we are confident that the survey methods and data processing techniques associated with the in situ GPS survey provide internally consistent ground-based results. While it is hard to isolate or quantify the non-zero errors associated with the ground-based GPS elevation data, we assume that the 0.055 to 0.059 m range of standard deviations is representative of the contribution of all of the terms mentioned in the previous section. The residuals presented here compare well with similar results from Siegfried et al. (2011) based on a dual traverse on 18 June 2009; their residual, based on differential post-processed techniques, was 0.009 m. Siegfried et al. (2011) also point out that the nearest-neighbor approach introduces new errors sources, and thus refrain from further interpretation, such as precision estimates.

Table 2 lists elevation bias and surface measurement precision relative to ground-based GPS survey data (i.e., lidar elevations – GPS elevations) for ATM and LVIS. The table lists results for both the nearest-neighbor and zone analysis. Further, the table presents two methods using DGPS post-processing techniques and one using the PPP method of post-processing.

The surface measurement precisions in Table 2, for both ATM and LVIS, are all less than 0.09 m and ranged from 0.039 to 0.087 m. The surface measurement biases in Table 2, for both ATM and LVIS, are all less than 0.12 m, with all of the IceBridge-related data collections (2009-2016) having measurement biases that range from -0.108 to +0.067 m. The overall largest measurement bias is associated with the 2007 LVIS airborne campaign, which was collected before the advent of IceBridge and as such did not undergo the comprehensive instrument calibration procedures now employed on IceBridge flights. The -0.108 m difference between the 2016 ATM and PPP GPS surface elevations is slightly larger than the other ground-based and airborne comparisons. During the ground survey, severe ionospheric activity had an impact on both the roving and base station GPS receivers for a period of 5 minutes. The resulting cycle slips were manually corrected in the GITAR DGPS processing, but not in the PPP processing, which could explain the better agreement between the GITAR and ATM comparison relative to the PPP and ATM comparison.

For both the nearest-neighbor and the zone analyses, $N$ from Equation 2 was generally consistent for the airborne lidars considered here. For ATM, $N$ for both the nearest-neighbor and zone analyses ranged from 220 to 494 per campaign, with an average of 351; for LVIS, $N$ for both the nearest-neighbor and zone analyses, ranged from 497 to 1219 per campaign, with an average of 858. For the zone analyses, the average number of GPS data points within the ATM 1 m diameter zone, or
search radius, ranged from 3 to 193 per campaign, with the mean being 27. The average number of GPS data points within the LVIS 10 m diameter zone ranged from 33 to 575 per campaign, with the mean being 304.

Our analysis indicates that there were no significant differences between results associated with the nearest-neighbor and zone methods of comparing the ground-based GPS and altimetry surface elevations (Table 2). The zone method may mitigate the impact of spurious outliers that could affect the surface measurement precision; this is potentially evident in a
comparison of the nearest-neighbor and zone results for the LVIS data, where precisions systematically improve slightly using the zone method. However, we note that results associated with a median method were all within 1 cm, and generally less than 0.1 cm, of results from the mean method. Thus, we consider the effects of outliers in this analysis to be negligible. Overall, the zone and nearest neighbor methods display similar results, most likely due to the relatively flat surface at Summit Station. Based on the ATM Level 2 ICESSN data product (Krabill, 2010) for all 3 passes associated with the 10
April 2014 flight, the slope over the traverse in the along-track direction is 0° and there is a gentle (0.1°) slope in the across-track direction (sloping toward the west); the difference between the maximum and minimum elevations in the vicinity of the traverse, based on the same data product, is only 1.05 m. Given these low-slope values, a geolocation error of 10 m is required to achieve a slope-induced elevation error of 0.01 m.

For this application, there are only small differences between the results associated with DGPS and PPP post-processing
methods associated with the ground-based GPS surveys. Results from each GPS processing method are statistically indistinguishable from one another and do not display a systematic pattern over the 8 observational periods that included both DGPS and PPP processing techniques. The similarity in relative bias between DGPS and PPP processing techniques is encouraging as there may be times when base station GPS data are unavailable for DGPS post-processing. Table 2, and the comparison of the residuals associated with the 5 May 2009 ground-based GPS data, suggest that, for this application, results
using PPP methods can be used to derive results that are as accurate and precise as those derived using DGPS methods for this small-scale ground-based GPS survey. We attribute some of the success of the PPP method to the ground-survey duration, which is sufficient to minimize errors associated with the convergence period (Bisnath and Gao, 2009), but short enough to minimize errors associated with the tropospheric modeling.

## 5 Discussion

The Summit Station ground-based GPS survey methods and data post-processing techniques are appropriate for airborne data validation. The three methods of data post-processing are internally consistent based on the difference between the two separate GPS surveys conducted on 5 May 2009. Further, for this application, there are only small differences between the

results associated with DGPS and PPP post-processing methods. The results presented here suggest that, given only roving-receiver GPS data, ground-based surface elevation data are still sufficiently accurate and precise for airborne elevation data validation.

Airborne and ground-based surveys should be coordinated with respect to timing. Two-thirds of the airborne lidar campaigns discussed here were within 2 days of the ground-based survey, which is a testament of the coordination between the airborne and ground-based teams. While we are limited with respect to observations, and we cannot state with certainty that the 20-day timing offset between the airborne and ground-based surveys was the unique source for the relatively poorer quality (0.09 m) surface measurement precision of the wide-scan lidar data for the 09 Apr 2015 flight, any elevation differences derived from environmental factors (e.g., snow, melt, or wind events) can be easily mitigated by closely coordinating the ground-based and airborne surveys.

Results for ATM and LVIS at Summit Station associated with the IceBridge campaigns date back to 2009 and provide an understanding and characterization of how these instruments perform and how that performance may evolve over time (Table 2, Fig. 4). ICESat-2 is scheduled to launch in 2018 and has a 3-year mission requirement; thus, for ICESat-2 post-launch validation activities that will utilize airborne sensors, it is essential to identify instruments now that are well characterized and well understood with respect to both accuracy and precision and to develop standardized survey, processing, and analytical techniques to ensure meaningful satellite data validation and interpretation.

As stated in the introduction, ICESat-2 mission requirements include the determination of ice-sheet elevation-change rates to an accuracy of less than or equal to 0.004 m $a^{-1}$ (Markus et al., 2016). This stringent requirement can only be met through statistical analysis of ICESat-2 elevation data at satellite ground-track crossovers. The ICESat-2 ice-sheet elevations released on the data product will be validated to 0.025 m. Given this requirement, and the ~0.7 m along-track sampling interval of ICESat-2, long length-scales (1000's of km) of airborne data over the ice sheets will be required for satellite data validation in order to increase the number of realizations of the satellite to airborne comparisons in order to significantly improve precision estimates based purely on an increased sample size (Boas, 1983). Therefore, ICESat-2 ice-sheet elevations will be validated using long length-scales of well-characterized airborne elevation data.

Results presented here are limited with respect to applicability to the entire ice sheet. Near the ice-sheet margins, airborne and satellite laser altimetry data are compromised due to increased surface roughness and slope, among other environmental variables (Brunt et al., 2010; 2014). However, the ground-based GPS elevation data collected near Summit Station provides a means to characterize airborne elevation data of ATM and LVIS. Comparisons between ATM and LVIS elevations and the ground-based elevations constrain the errors of the airborne datasets. Thus, in situ data, even on short length-scales and over flat surfaces, can form part of a strategy to validate data from airborne, and ultimately satellite, platforms. Further, the Summit Station survey has been conducted monthly since August 2006. The long, dense time series associated with the ground-based survey will ultimately provide the seasonal information required to derive meaningful surface-elevation-change interpretation of ICESat-2 data. This reinforces the importance of long-duration, high-frequency, ground-based observations in linking in situ, airborne, and satellite observations.

## 6 Conclusions

It is often difficult to collect sufficient length scales of in situ elevation data to provide meaningful statistics for satellite laser altimetry validation. Therefore, a nested approach for validation of satellite elevation is commonly employed. In a nested approach, ground-based GPS data are collected to constrain the elevation bias and measurement precision of the airborne lidar data. Airborne surveys can then be designed and conducted on longer length-scales to provide the amount of airborne data required to make statistically meaningful assessments of satellite elevation accuracy and precision.

We have presented a comparison of airborne lidar data with in situ GPS data, over relatively flat terrain associated with the ice-sheet interior, in preparation for validation efforts associated with ICESat-2. Results were consistent given various data processing methods (PPP and DGPS) and data analysis methods (nearest-neighbor or zone analysis). The 11 km Summit Station ground-based GPS survey intersects just 6 km of the satellite ground track (Fig. 1). Therefore, to make statistically robust assessments of ICESat-2 elevations, a nested approach will need to be employed for data validation. However, the Summit Station survey provides a means to characterize airborne instruments, which will in turn collect sufficient amounts of data required for satellite data validation. Further, results presented here date back to 2007, providing a characterization of how airborne instrument performance may evolve over time. For ATM our analysis spans four generations of instrument and data systems (Table 2) documenting long-term data consistency and accuracy. Long-term data consistency will be crucial for producing a cross-calibrated and validated surface elevation change time series using ICESat and ICESat-2 data. From this comparison of airborne and ground based data collected under standardized protocols, we find that both ATM and LVIS are sufficiently characterized and thus well poised to be integrated with an ICESat-2 data validation strategy.

## 7 Data availability

Summit ground-based GPS data associated with the airborne lidar data are available online, as the supplement related to this article (doi:10.5194/tc-2016-214-supplement). The base station GPS data are publicly available on the UNAVCO Data Archive Interface (http://www.unavco.org/data/gps-gnss/data-access-methods/dai2/app/dai2.html) and are included in the supplement related to this article (doi:10.5194/tc-2016-214-supplement). NASA ATM and the LVIS (2010) data are publicly available on the NSIDC Operation IceBridge Data Portal (http://nsidc.org/icebridge/portal/). The Pre-IceBridge LVIS data (2007) are also publicly available at the NSIDC (https://nsidc.org/data/blvis2). MABEL lidar data are publicly available on the NASA ICESat-2 data page (http://icesat-2.gsfc.nasa.gov/icesat2/data/mabel/mabel_docs.php). The NASA GSFC surface-finding algorithm is available from the authors upon request (kelly.m.brunt@nasa.gov).

## Acknowledgements

We thank the NASA ICESat-2 Project Science Office for funding this data analysis and for MABEL data collection, processing, and distribution. Further, we thank the NASA Armstrong Air Operations Facility for MABEL data collection

(specifically pilots T. Williams and D. S. Broce). We thank Operation IceBridge for the data collection and processing associated with the ATM and LVIS airborne components of this project. We thank the National Science Foundation and the Summit Station Science Coordination Office (SCO) (NSF PLR 1042358) for support for the ground-based field component of this project. Further, this project would not have been possible without the work of many Summit Station Science

Technicians, who collected the in-situ GPS data. We thank K. Krabill (NASA GSFC WFF), C. Brooks, and D. Rabine (NASA GSFC) for GPS support. We thank the National Snow and Ice Data Center (NSIDC) for IceBridge data distribution. Finally, we thank our editor (Etienne Berthier) and 2 anonymous reviewers for insightful and constructive comments to earlier drafts of this manuscript.

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

**Table 1: Airborne laser altimetry and ground-based GPS survey dates and comments.**

| Lidar | Survey altitude | Lidar survey date | GPS survey date | Offset (days) | $h_{AntPost}$ (m) | $h_{RunnerDepth}$ (m) | Comments |
|---|---|---|---|---|---|---|---|
| ATM [2] | 450 m AGL | 05 May 2009 | 05 May 2009 | 0 | 1.785 | 0.0200 | 2 GPS surveys; OIB |
| ATM | 450 m AGL | 11 Apr 2012 | 11 Apr 2012 | 0 | 1.785 | 0.0175 | OIB |
| ATM | 450 m AGL | 10 Apr 2014 | 02 Apr 2014 | -8 | 1.797 | 0.0175 | 3 ATM passes; OIB |
| ATM | 450 m AGL | 02 May 2014 | 13 May 2014 | +11 | 1.797 | 0.0150 | OIB |
| ATM [A, *] | 450 m AGL | 09 Apr 2015 | 29 Apr 2015 | +20 | 1.797 | 0.0125 | OIB |
| ATM [B] | 450 m AGL | 19 May 2016 | 19 May 2016 | 0 | 1.797 | 0.0350 | OIB |
| LVIS [1] | ~4600 m AGL | 20 Sept 2007 | 18 Sept 2007 | -2 | 1.785 | 0.0175 | 2 LVIS passes; Pre-OIB |
| LVIS [1, C] | ~4600 m AGL | 14 Apr 2010 | 14 Apr 2010 | 0 | 1.785 | 0.0150 | OIB |
| MABEL [D] | ~16,000 m AGL | 08 Apr 2012 | 08 Apr 2012 | 0 | 1.785 | 0.0150 | 3 MABEL passes; IS-2 |
| MABEL [D] | ~16,000 m AGL | 12 Apr 2012 | 11 Apr 2012 | -1 | 1.785 | 0.0175 | 2 MABEL passes; IS-2 |

[1] Indicates ITRF00 as the airborne data post-processing datum; [2] indicates ITRF05; all others are ITRF08.
[A] Indicates that the instrument was integrated with a C-130; [B] indicates that the instrument was integrated with the NOAA P-3; [C] indicates that the instrument was integrated with the NASA DC-8; [D] indicates that the instrument was integrated with the NASA ER-2; all others surveys were flown on the NASA P-3.
[*] Indicates that both narrow- and wide-scanning lidar data are available; all other ATM analysis is associated solely with wide-scan lidar data.
OIB in the 'Comments' field indicates that data are from IceBridge and that they are available via the NSIDC Operation IceBridge Data Portal (http://nsidc.org/icebridge/portal).
Pre-OIB in the 'Comments' field indicates that data are pre-IceBridge; they are available via NSIDC (http://nsidc.org/data/blvis2).
IS-2 in the 'Comments' field indicates that data are available via the ICESat-2 website (http://icesat-2.gsfc.nasa.gov).

**Table 2: Airborne lidar elevation bias and surface measurement precision (in m) relative to ground-based GPS survey data (i.e., lidar elevations – GPS elevations) using 'nearest-neighbor' and 'zone' analysis.**

| Lidar Survey | Lidar Version (scan angle) | DGPS1 bias ± precision: nearest-neighbor (m) zone (m) | DGPS2 bias ± precision: nearest-neighbor (m) zone (m) | PPP bias ± precision: nearest-neighbor (m) zone (m) |
|---|---|---|---|---|
| ATM 05 May 2009 [2] | 4B/T2 (30°) | 0.055 ±0.074; $N$ = 255<br>0.055 ±0.074; $N$ = 255 | 0.005 ±0.073; $N$ = 254<br>0.005 ±0.074; $N$ = 254 | -0.026 ±0.075; $N$ = 253<br>-0.026 ±0.075; $N$ = 253 |
| ATM 11 Apr 2012 | 4B/T4 (30°) | 0.067 ±0.045; $N$ = 320<br>0.067 ±0.045; $N$ = 320 | -0.014 ±0.055; $N$ = 323<br>-0.014 ±0.055; $N$ = 323 | 0.008 ±0.039; $N$ = 321<br>0.008 ±0.039; $N$ = 321 |
| ATM 10 Apr 2014 | 4B/T4 (30°) | 0.018 ±0.076; $N$ = 491<br>0.018 ±0.076; $N$ = 491 | 0.040 ±0.077; $N$ = 491<br>0.040 ±0.077; $N$ = 491 | -0.021 ±0.075; $N$ = 494<br>-0.021 ±0.075; $N$ = 494 |
| ATM 02 May 2014 | 4B/T4 (30°) | 0.005 ±0.054; $N$ = 220<br>0.005 ±0.054; $N$ = 220 | 0.037 ±0.051; $N$ = 221<br>0.037 ±0.051; $N$ = 221 | -0.005 ±0.052; $N$ = 223<br>-0.005 ±0.052; $N$ = 223 |
| ATM 09 Apr 2015 [A] | 5A/T3 (30°) | 0.004 ±0.088; $N$ = 470<br>0.004 ±0.088; $N$ = 470 | -0.026 ±0.087; $N$ = 476<br>-0.026 ±0.087; $N$ = 476 | -0.064 ±0.087; $N$ = 472<br>-0.064 ±0.087; $N$ = 472 |
| ATM 09 Apr 2015 [A] | 5B/T5 (5°) | 0.043 ±0.068; $N$ = 365<br>0.043 ±0.068; $N$ = 365 | 0.015 ±0.070; $N$ = 366<br>0.015 ±0.070; $N$ = 366 | -0.021 ±0.068; $N$ = 368<br>-0.021 ±0.068; $N$ = 368 |
| ATM 19 Apr 2016 [B] | 5A/T2 (30°) | -0.070 ±0.075; $N$ = 331<br>-0.070 ±0.075; $N$ = 331 | -0.043 ±0.073; $N$ = 329<br>-0.043 ±0.072; $N$ = 329 | -0.108 ±0.059; $N$ = 336<br>-0.108 ±0.059; $N$ = 336 |
| LVIS 20 Sept 2007 [1] | Pre-OIB | 0.115 ±0.061; $N$ = 1219<br>0.116 ±0.057; $N$ = 1219 | 0.085 ±0.059; $N$ = 1219<br>0.086 ±0.056; $N$ = 1219 | 0.081 ±0.061; $N$ = 1218<br>0.082 ±0.057; $N$ = 1218 |
| LVIS 14 Apr 2010 [1, C] | OIB | 0.037 ±0.064; $N$ = 497<br>0.035 ±0.060; $N$ = 497 | -0.004 ±0.064; $N$ = 497<br>-0.006 ±0.060; $N$ = 497 | -0.024 ±0.061; $N$ = 497<br>-0.027 ±0.058; $N$ = 497 |

Elevation bias (lidar elevation – GPS elevation) and surface measurement precision are in m. DGPS1 is TRACK software processing results, processed to the L1 phase center, 0.056 m above the base plane; DGPS2 is GITAR software processing results, processed to the LC phase center, 0.061 m above the base plane; PPP is Inertial Explorer software processing results, processed to the L1 phase center, 0.056 m above the base plane.
[1] Indicates ITRF00 as the post-processing datum; [2] indicates ITRF05; all others are ITRF08.
[A] Indicates that the instrument was integrated with a C-130; [B] indicates that the instrument was integrated with the NOAA P-3; [C] indicates that the instrument was integrated with the NASA DC-8; all others surveys were flown on the NASA P-3.

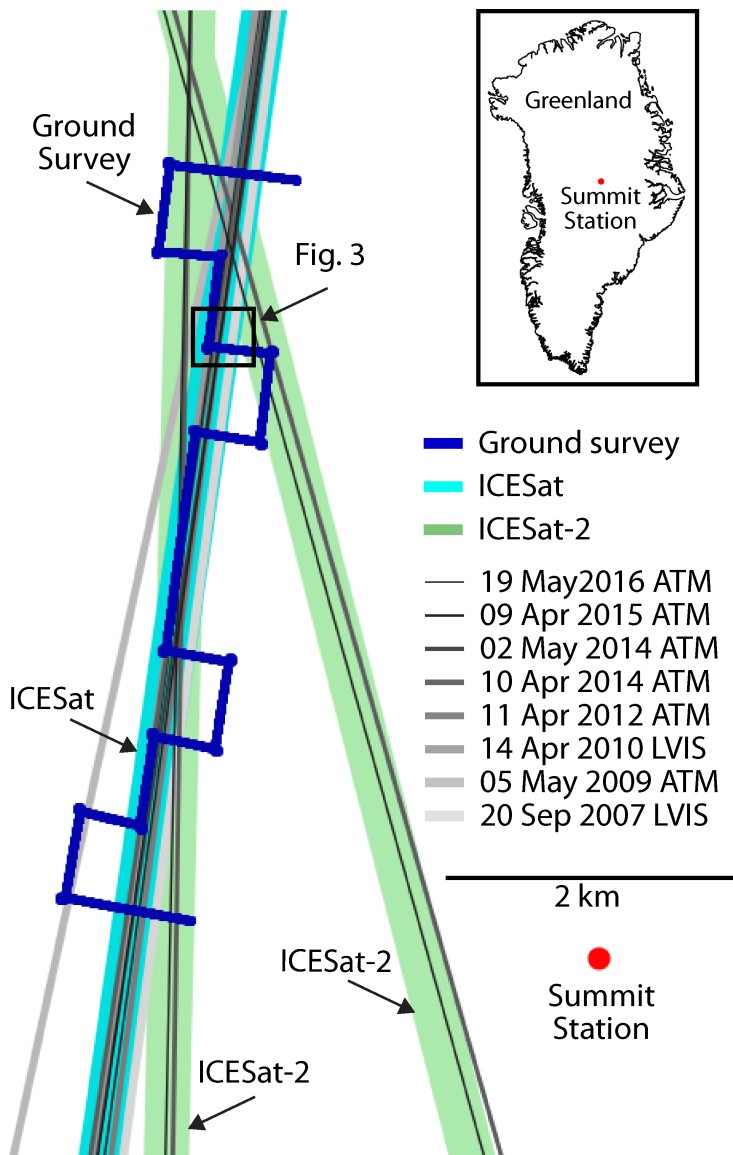

**Figure 1:** Map of the Summit Station area, including: representative ground-based GPS survey line (blue line); airborne lidar surveys (gray lines, thicker lines occur earlier); ICESat ground track #0412 (cyan line); ICESat-2 ground tracks (green lines); and Summit Station (red dot). The distance between Summit Station and the southern end of the traverse is ~3 km; the distance between Summit Station and the northern end of the traverse is ~6.5 km. Relative to the trend of ICESat track #0412, the ground-based GPS survey line is oriented both along-track and across-track in order to better characterize the surface slope.

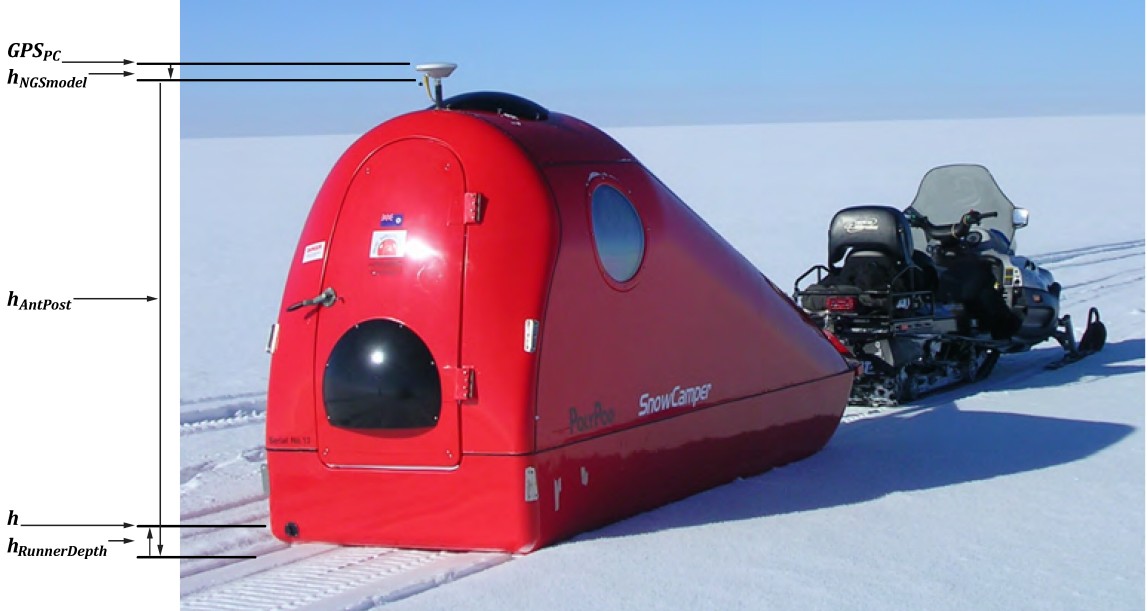

**Figure 2:** The roving GPS antenna, sled, and snowmobile configuration. $GPS_{PC}$ is the surveyed position solution to the phase center of the antenna, $h_{NGSmodel}$ is the NGS model distance between the antenna phase center and the antenna base plane, $h_{AntPost}$ is the height of the antenna post (Table 1), $h_{RunnerDepth}$ is the depth of the sled runners in the snow surface (Table 1), and $h$ is the snow surface (Equation 1).

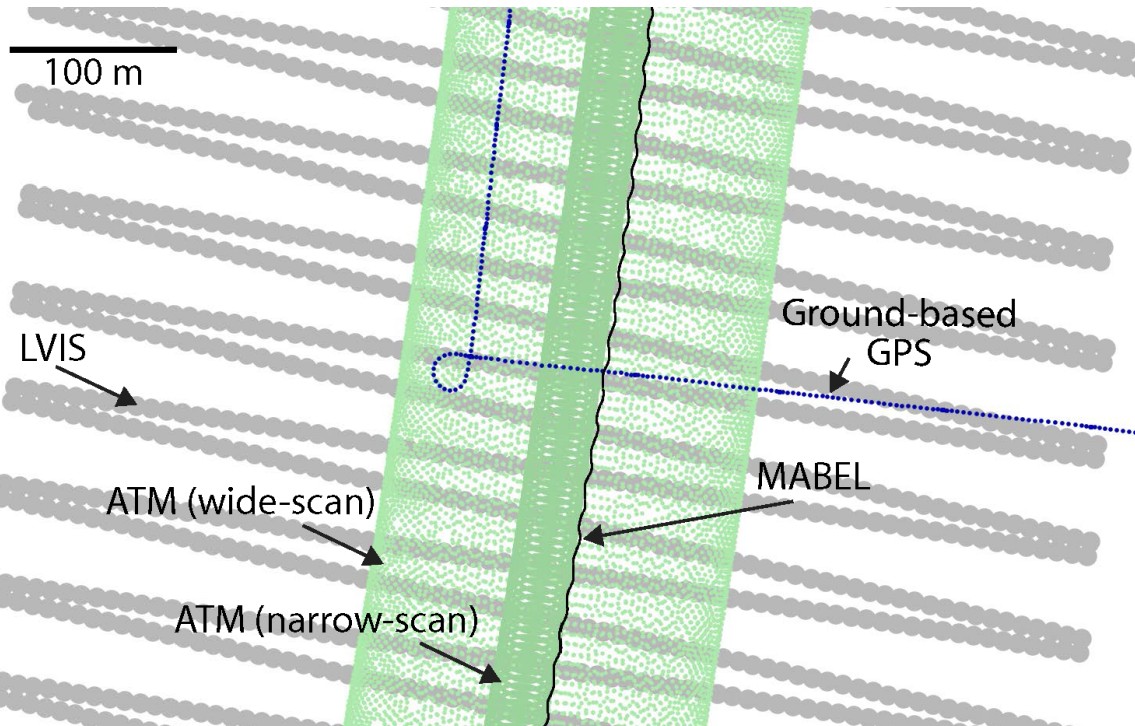

**Figure 3:** Schematic representation of the lidar measurement strategies and ground-based GPS sample spacing. LVIS measurements (gray): ~10 m diameter footprint and a 1000 m across-track swath width. ATM measurements (green): ~1 m diameter footprint and either a 40 m (narrow-scan; post 2012) or a 250 m (wide-scan) across-track swath width. Ground-based GPS data (blue points) indicate sample spacing. MABEL measurements (black dots) are included to illustrate the limitations of a profiling lidar for this application.

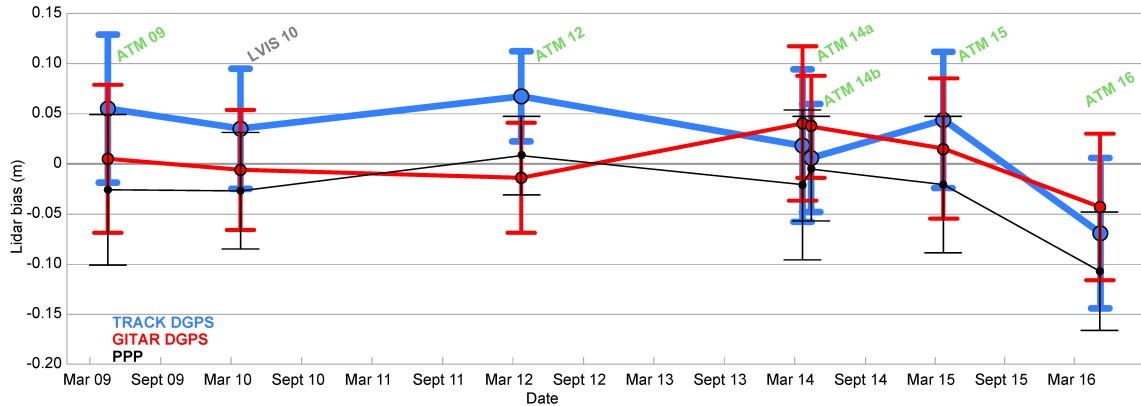

**Figure 4:** Performance of ATM and LVIS over Summit Station through time. Date versus lidar surface bias (m), for ATM and LVIS, for the IceBridge campaigns. Error bars represent surface measurement precision. TRACK DGPS (blue), GITAR DGPS (red), and Inertial Explorer PPP (black) GPS post-processing results are presented. ATM 15 represents the narrow-scan lidar system only. We note that for the GITAR solutions, the base station positions represent an average of four days of data; while for the TRACK solutions, the base station positions represent an average recorded over the duration of the ground-based survey. This may account for the slight offset associated with the TRACK solutions.

