# Peer review of "Assessment of NASA airborne laser altimetry data using groundbased GPS data near Summit Station, Greenland"

_The Cryosphere, 2016_

## Referee Comment (RC1) · Anonymous Referee #1 · 15 Nov 2016

I think the paper makes a significant contribution to our knowledge of assessment of altimetry data using groundbased GPS data. I believe the paper should be published after minor revision.

Minor comments: Page 2, line 14, concerning satellite-derived time series. Add references to other than Zwally, 2005. Page 3, line 11 and 18 (and throughout the paper). 11,000 m and 6,000 m, change to 11 km and 6 km. I Assume the track is not exactly 11,000 m ? Page 3 line 18: After the text "...long and dense in situ observation of ice-sheet elevation change". Perhaps add reference to other papers combining GPS and ICESat, ATM e.g. Larsen, S. H., et al, J. Geophys. Res. Earth Surf., 121, 241–256, doi:10.1002/2015JF003507.

[Figure]

Page 3, line 24. Is Trimble R7 receiver and Zephyr antenna used for all years ? Page 3, line 26. If possible add a photo of the antenna mounted on a static metal post on the sled. Page 6, line 18-31. I find this section confusing. You have listed names of several software packages used to process GPS data, however, no useful information about how the GPS data is processed is provided. How do you correct for troposphere delay, ionosphere? Which model is used? what cut-off elevation angle do you use? how do you deal with multipath etc. . . .? It is also important that same GPS clock/orbit products are used by the different software packages, otherwise you may add an extra bias to your GPS solutions.

Discussion: Potential future work. In addition to surface elevations, you could compare elevation change rates. Here, you could take advantage of the continuously operating GPS base station at Summit Station. The reflected GPS signals from the summit station, can be used to measure GPS reflected height of the surface (see Larson et al, 2015, Journal of Glaciology, Vol. 61, No. 225, 2015 doi: 10.3189/2015JoG14J130).

---

## Referee Comment (RC2) · Anonymous Referee #2 · 5 Dec 2016

This is a short and concise paper on the quality of airborne laser altimetry data over a flat ice sheet surface. The authors calculate error statistics from comparisons with near-coincident surface GPS profiling near the Greenland Summit Station. Considering how often these data are used in ice-sheet change assessments, and how accurate they need to be to detect cm-level elevation changes, I think it is a timely and highly appreciated contribution to the community. It also paves the way for using designated airborne surveys to validate satellite altimetry data, in particular the upcoming ICESat-2 mission.

I have only some smaller comments and questions as given in chronological order below. They all refer to line numbers in the discussion paper, but some are of more

general character and could warrant changes also elsewhere in the manuscript.

P1, L21: I think it's worth to mention that you get equally good correspondence with DGPS and PPP techniques. The latter could simplify fieldwork for many applications.

P2, L7: I would also cite Borsa et al. (2014, The Cryosphere) here since the other papers are prior to that do not all account for the Gaussian-Centroid bias.

P2, L18: I don't think airplanes can really bridge the gap between satellite missions at the scale of ice sheets, so I would add "... in areas of special interest" or something like that.

P2, L31: Write out GEDI.

P4, L20: I don't see the need for this abbreviation since it is only used a few times.

P5, L1: The software incorporates GLONASS, but does any of the actual observations include that? It would be a strength if they did, and in that case you should use the general term GNSS in cases where you do not mean solely the GPS system.

P5, L21: Is GLONASS or GALILEO included in any of this processing? If so, it should be mentioned.

P5, L22: I don't think the term PPP has been introduced yet.

P5, L23: Since MABEL is included for reference, I think it's also worth to describe ICESat-2 in a similar fashion as a part of the same section or a brief separate one. In perspective of future ICESat-2 validation, it would be useful to know roughly how many comparison points one would get with the present GPS survey lines.

P6, L23: Since this PPP software is commercial and many people these days use freely available services like the Canadian CSRS-PPP, it would be nice to see how one of these automatic processors would compare in the validation exercise.

P6, L28: I miss some small details on the processing: Were final IGS orbits used

in all processing cases? Same for clock corrections? How were tropospheric and ionospheric errors dealt with? Was a cut-off angle used for satellite elevation to mitigate multipath?

P7, L4: How were these solid earth tides estimated?

P7, L16: I don't understand this logic. From these numbers I only gather a footprint spacing of 5 m, not the actual size.

P8, L5-12: While uncertainty in the ground-based GPS probably influences the inferred lidar precision, it is also worth to mention that the two surface measurement techniques are partly correlated through their common use of GPS (and partly processing techniques) for vehicle positioning. I don't think this will have a large impact, but it is worth to discuss briefly. The problem could be mitigated by additional or isolated use of GLONASS or GALILEO in one of the platforms, but that might not be possible.

P9, L16: In case of outliers it would make most sense to use the median value in each zone. Did you also try that? Worth to mention whether or not it makes a difference.

P9, L18: How flat is 'relatively flat'? It would be good to provide some kind of information about the summit topography, for example elevation range, mean slope, or average elevation impact of a given geolocation error like 5-10 m.

P9, L27: This is an interesting finding that I think should also be mentioned in the abstract or conclusions.

P11, L16: Credits to the authors for making all data easily available. Exemplary!

Fig 3: The TRACK solutions seem to infer a higher lidar bias than GITAR and PPP. Is this random or could there be a viable explanation related to processing?

---

## Author Response (AR1)

Dear Etienne-

Thank you for accepting the role of Editor for our manuscript (Assessment of NASA airborne laser altimetry data using ground-based GPS data near Summit Station, Greenland). I greatly appreciate the time that you have already put into this project.

I have uploaded a point-by-point response to the 2 anonymous reviews and to the comments that you provided as editor. All of these comments substantially improved this manuscript.

Thank you also for the extension. What this time allowed us to do was 1) address a processing glitch that had previously prevented us from DGPS post-processing the 2014 and 2015 data; and 2) compare previously unavailable 2016 ATM data to ground-based data. Therefore, Table 2, which previously had a few 'Rover only' comments, as opposed to results, has been completely filled in and is current through 2016.

Thank you again for your constructive feedback and the time that you have already invested in this manuscript.

Respectfully,
Kelly

**Response to Anonymous Referee #1:**

We thank this anonymous reviewer for their constructive comments. Edits based on your input (and that of the second reviewer) have substantially improved this manuscript.

Here are our responses (in red) to your specific comments (in black):

I think the paper makes a significant contribution to our knowledge of assessment of altimetry data using groundbased GPS data. I believe the paper should be published after minor revision.

Minor comments:
Page 2, line 14, concerning satellite-derived time series. Add references to other than Zwally, 2005.
This was an oversight. We have added Velicogna et al., 2014, Shepherd et al., 2012, and Zwally et al., 2011; we feel that these references cover various satellite methods for both Greenland and Antarctica.

Page 3, line 11 and 18 (and throughout the paper). 11,000 m and 6,000 m, change to 11 km and 6 km. I Assume the track is not exactly 11,000 m ?
We had originally used 'm' throughout for consistency, but the reviewer makes an excellent point. We have changed most references to 1000 – 10,000 m to km. This also includes the caption and scale bar of Figure 1.

Page 3 line 18: After the text "…long and dense in situ observation of icesheet elevation change". Perhaps add reference to other papers combining GPS and ICESat, ATM e.g. Larsen, S. H., et al, J. Geophys. Res. Earth Surf., 121, 241–256, doi:10.1002/2015JF003507.
Our text here was intended to highlight the very unique nature of this in situ dataset, which is a 10-year, monthly, ground-based traverse, that continues to this day (and is expected to continue through the ICESat-2 mission, into the 2020's); thus, this should prove to be a roughly 15-year dataset. We are reluctant to diminish this point with a comparison to a 5-year in situ record, at 4 sites (not a survey line). But to address this point, we have added text that reinforces the uniqueness of the dataset.

Page 3, line 24. Is Trimble R7 receiver and Zephyr antenna used for all years?
Yes, the same equipment was used all 4 years. We have added a line that explicitly states this: (" *… ; we note that the kinematic surveys have always been conducted using this equipment.*"). Further, we added a similar statement about the base station: ("*For the duration of the survey time series, the base station has been a dual-frequency Trimble NetRS receiver recording at 1 Hz with a Trimble Zephyr Geodetic antenna…*").

Page 3, line 26. If possible add a photo of the antenna mounted on a static metal post on the sled.
This is an excellent suggestion. We have added a new Figure 2, which not only provides a photo of the setup, but also gives approximate numbers for the corrections from the GPS phase center back to the snow surface.

Page 6, line 18-31. I find this section confusing. You have listed names of several software packages used to process GPS data, however, no useful information about how the GPS data is processed is provided. How do you correct for troposphere delay, ionosphere? Which model is used? what cut-off elevation angle do you use? how do you deal with multipath etc…? It is also important that same GPS clock/orbit products are used by the different software packages, otherwise you may add an extra bias to your GPS solutions.
We have added a lengthy paragraph with the details of each of the processing methods: ("*Independent of post-processing method, all of the ground-based GPS solutions are based on final precise orbit and clock information from the Crustal Dynamics Data Information System, or CDDIS, at GSFC. Processing using TRACK corrected for errors associated with the ionosphere by incorporating an IGS data product. To mitigate the effect of multipath distortion, all processing methods used a cut-off angle (7.5°, 10°, and 12° for Inertial Explorer, TRACK and GITAR, respectively). Inertial Explorer and TRACK used a Saastamoinen model to correct for tropospheric delay, while GITAR used a gridded reanalysis data product from the National Centers for Environmental Prediction (NCEP). And all processing methods corrected for solid Earth tides based on an Earth Rotation and Reference System Services, or IERS, model.*")

Discussion: Potential future work. In addition to surface elevations, you could compare elevation change rates. Here, you could take advantage of the continuously operating GPS base station at Summit Station. The reflected GPS signals from the summit station, can be used to measure GPS reflected height of the surface (see Larson et al, 2015, Journal of Glaciology, Vol. 61, No. 225, 2015 doi: 10.3189/2015JoG14J130).
The idea of investigating change rates is great, and it is exciting that these data are available for such a study. However, at this time this is outside of the scope of this paper, where we wanted to stay focused on aircraft/satellite applications.

**Response to Anonymous Referee #2:**

We thank this anonymous reviewer for their constructive comments. Edits based on your input (and that of the first reviewer) have substantially improved this manuscript.

Here are our responses (in red) to your specific comments (in black):

This is a short and concise paper on the quality of airborne laser altimetry data over a flat ice sheet surface. The authors calculate error statistics from comparisons with near-coincident surface GPS profiling near the Greenland Summit Station. Considering how often these data are used in ice-sheet change assessments, and how accurate they need to be to detect cm-level elevation changes, I think it is a timely and highly appreciated contribution to the community. It also paves the way for using designated airborne surveys to validate satellite altimetry data, in particular the upcoming ICESat-2 mission.

I have only some smaller comments and questions as given in chronological order below. They all refer to line numbers in the discussion paper, but some are of more general character and could warrant changes also elsewhere in the manuscript.

P1, L21: I think it's worth to mention that you get equally good correspondence with DGPS and PPP techniques. The latter could simplify fieldwork for many applications. We were pleasantly surprised by that result, with respect to this application. We have added text to the abstract ("*Ground-based GPS positions determined both with and without differential post-processing techniques provided consistent solutions.*").

P2, L7: I would also cite Borsa et al. (2014, The Cryosphere) here since the other papers are prior to that do not all account for the Gaussian-Centroid bias. Excellent point; we have added this to the manuscript.

P2, L18: I don't think airplanes can really bridge the gap between satellite missions at the scale of ice sheets, so I would add "...in areas of special interest" or something like that. Agreed. We have modified the text to capture your comment, while still acknowledging the OIB mission: '… *1) "bridging the data gap" between the satellite missions, with a focus on areas of significant change and interest.*'

P2, L31: Write out GEDI. Thanks for catching this; we have added the full name.

P4, L20: I don't see the need for this abbreviation since it is only used a few times. This acronym does show up 2 other times in the text and multiple times in Table 1. So we are inclined to keep it. Further, other publications associated with NASA altimetry (especially when the altimeter is integrated with the ER-2) use ASL, and thus our explicit use of AGL in this manuscript is for clarity.

P5, L1: The software incorporates GLONASS, but does any of the actual observations include that? It would be a strength if they did, and in that case you should use the general term GNSS in cases where you do not mean solely the GPS system.

The reviewer makes a good point here about non-GPS constellations. The Summit Base Station and the roving receiver associated with the traverse only logged data from the GPS constellation. We have added text to clarify this ("*Both the base station and the rover logged data solely from the GPS constellation.*").

P5, L21: Is GLONASS or GALILEO included in any of this processing? If so, it should be mentioned.

ATM has used GLONASS since 2011; we have added that detail to the text. LVIS (which is what this comment was addressing) uses GPS only; we have added text to clarify this ("*Position information for the 2007 and 2010 LVIS campaigns incorporated data from the GPS constellation only.*").

P5, L22: I don't think the term PPP has been introduced yet.

Excellent catch; it had not been defined. Further, we moved the succinct definitions of both 'DGPS' and 'PPP' methods up to their first usage (originally they were defined at the start of section 3).

P5, L23: Since MABEL is included for reference, I think it's also worth to describe ICESat-2 in a similar fashion as a part of the same section or a brief separate one. In perspective of future ICESat-2 validation, it would be useful to know roughly how many comparison points one would get with the present GPS survey lines.

This is a good point, although we don't feel that this is the place for this comment. We note that the pertinent aspects of ICESat-2/ATLAS (e.g., along-track sample spacing of 70 cm) are described in the Introduction. We now recall this information at the end of the Discussion and add the text: ("*The ICESat-2 ice-sheet elevations released on the data product will be validated to 0.025 m. Given this requirement, and the ~0.7 m along-track sampling interval of ICESat-2, long length-scales (1000's of km) of airborne data over the ice sheets will be required for satellite data validation in order to increase the number of realizations of the satellite to airborne comparisons in order to significantly improve precision estimates based purely on an increased sample size (Boas, 1983).*").

P6, L23: Since this PPP software is commercial and many people these days use freely available services like the Canadian CSRS-PPP, it would be nice to see how one of these automatic processors would compare in the validation exercise.

The reviewer makes a great point. We experimented with an online service (specifically JPL's GIPSY) and had mixed results. We attribute much of this to the handling of the atmospheric corrections. These black-box resources are fantastic, especially for static applications. However, for cm-level accuracy associated with kinematic surveys, and for direct comparison to the aircraft data (i.e., specifying ITRF00, 05, or 08), we had more success with software that we could customize/control for this application. ***We also note that there is variance in***

***online PPP services; casual users of these services need to use caution when interpreting results from these systems (this bold/italics statement is not in the online response).***

P6, L28: I miss some small details on the processing: Were final IGS orbits used in all processing cases? Same for clock corrections? How were tropospheric and ionospheric errors dealt with? Was a cut-off angle used for satellite elevation to mitigate multipath?

P7, L4: How were these solid earth tides estimated?

Reviewer 1 made a similar comment. We have added details about each of the 3 processing methods: "*Independent of post-processing method, all of the ground-based GPS solutions are based on final precise orbit and clock information from the Crustal Dynamics Data Information System, or CDDIS, at GSFC. Processing using TRACK corrected for errors associated with the ionosphere by incorporating an IGS data product. To mitigate the effect of multipath distortion, all processing methods used a cut-off angle (7.5°, 10°, and 12° for Inertial Explorer, TRACK and GITAR, respectively). Inertial Explorer and TRACK used a Saastamoinen model to correct for tropospheric delay, while GITAR used a gridded reanalysis data product from the National Centers for Environmental Prediction (NCEP). And all processing methods corrected for solid Earth tides based on an Earth Rotation and Reference System Services, or IERS, model.*"

P7, L16: I don't understand this logic. From these numbers I only gather a footprint spacing of 5 m, not the actual size.

You are correct. Our error here was associated with radius/diameter confusion. We have changed this text to read: "We note that the GPS data were collected at 1 Hz, with the snowmobile operating at ~5 m s$^{-1}$, giving the GPS data an effective 5 m diameter footprint."

P8, L5-12: While uncertainty in the ground-based GPS probably influences the inferred lidar precision, it is also worth to mention that the two surface measurement techniques are partly correlated through their common use of GPS (and partly processing techniques) for vehicle positioning. I don't think this will have a large impact, but it is worth to discuss briefly. The problem could be mitigated by additional or isolated use of GLONASS or GALILEO in one of the platforms, but that might not be possible.

This is a great point. We don't have the option of GLONASS or GALILEO for the existing time series of ground-based survey data. This might be something that could be incorporated in future logging during that traverse. LVIS and MABEL only use the GPS constellation. ATM has used both GPS and GLONASS since 2011 (the date is now noted in the text); but we suspect that the solutions would be compromised if we were to use solely GLONASS data. That being said, to acknowledge this comment, we have added a note that we are mindful of the correlation issue: (" *… and 5) observational errors such as variable penetration of the sled into the snow along the course of the survey. Further, we note that the existing*

*ground-based and airborne elevation data are partially correlated, as they are based on similar GPS measurement strategies."*).

P9, L16: In case of outliers it would make most sense to use the median value in each zone. Did you also try that? Worth to mention whether or not it makes a difference.
This is a fantastic point and a great addition to the manuscript. We have edited/added the following text to the manuscript: "*The zone method may mitigate the impact of spurious outliers that could affect the surface measurement precision; this is potentially evident in a comparison of the nearest-neighbor and zone results for the LVIS data, where precisions systematically improve slightly using the zone method. However, we note that results associated with a median method were all within 1 cm, and generally less than 0.1 cm, of results from the mean method. Thus, the effects of outliers in this analysis are generally negligible. Overall, the zone and nearest neighbor methods display similar results, most likely due to the relatively flat surface at Summit Station.*"

P9, L18: How flat is 'relatively flat'? It would be good to provide some kind of information about the summit topography, for example elevation range, mean slope, or average elevation impact of a given geolocation error like 5-10 m.
We assessed the across- and along-track slope based on an ATM Level 2 data product (both either 0° or near 0°) and also included the difference between the maximum and minimum elevations in this region and the impact at 10 m geolocation error (all good additions to the manuscript). We included this text and a new reference: ("*Based on the ATM Level 2 Icessn data product (Krabill, 2010) for all 3 passes associated with the 10 Apr 2014 flight, the slope over the traverse in the along-track direction is 0° and there is a gentle (0.1°) slope in the across-track direction (sloping toward the west); the difference between the maximum and minimum elevations in the vicinity of the traverse, based on the same data product, is only 1.05 m. Given these low slope values, a geolocation error of 10 m is required to achieve a slope-induced elevation error of 0.01 m.*")

P9, L27: This is an interesting finding that I think should also be mentioned in the abstract or conclusions.
We have added this to the abstract (see comment above) and now to the conclusions: "*Results were consistent given various data processing methods (PPP and DGPS) and data analysis methods (nearest-neighbor or zone analysis).*"

We do, however, want to be a little cautious about overstating this result. Our original text does point out that this is the case for 'this application'. We have now also added the following text at the very end of the 'Results' section: "*We attribute some of the success of the PPP method to our ground-survey duration, which is sufficient to minimize errors associated with the convergence period (Bisnath and Gao, 2009), but short enough to minimize errors associated with the tropospheric modeling.*"

P11, L16: Credits to the authors for making all data easily available. Exemplary!
Thank you. But this is probably more of a reflection on new requirement from The Cryosphere.

Fig 3: The TRACK solutions seem to infer a higher lidar bias than GITAR and PPP. Is this random or could there be a viable explanation related to processing?
We attribute this to the fact that the GITAR solutions use a base-station position that is averaged over 4 days, while for TRACK, the base station positions represent an average recorded over the duration of the ground-based survey (~3 to 4 hours). While we originally noted this in the text (in Section 3.1), we have now added similar text to the caption.

**Response to Editor's Comments:**

We thank the Editor for his constructive comments. Edits based on your input (and that of 2 anonymous reviewers) have substantially improved this manuscript.

Here are our responses (in red) to your specific comments (in black):

Significance (Impact):
This is maybe the weakness of the paper. Its scope is rather technical for TC with little new glaciological knowledge. On the other hand, I admit that the audience for this study is the ice sheet community (rather than the remote sensing community) given that such an assessment is expected for these widely-used airborne data. One aspect that in my view currently limits the significance of the study is that the error assessment is performed over a very flat area in the central part of the ice sheet which is extremely favourable for laser altimetry measurements. To what extent can the result be extended to other regions of the ice sheets (steeper, crevassed, etc…) and to Arctic glaciers? Something to probably discuss more extensively.

The paper is short and well-structured. The discussion is probably the weakest part. It could be improved by comparing the present results with previous works (if any, maybe on the Antarctic Ice Sheet?) and describing the limitation of the study.

These are fair points. One goal of this manuscript is to provide the ice-sheet community with a reference for ICESat-2/airborne analysis, in preparation for that satellite mission, which will be a phenomenal asset for the whole ice-sheet community. So while the paper is technical, we feel that it is germane to The Cryosphere community.

To address some of these comments, language was added to an early draft of the abstract stating that the results are for the ice-sheet interior (*"Surface elevation biases for these altimeters, over the flat, ice-sheet interior, are less than 0.12 m, while assessments of measurement precision are 0.09 m or better."*). Further, we added a similar reminder to the conclusions (*"We have presented a comparison of airborne lidar data with in situ GPS data, over relatively flat terrain associated with the ice-sheet interior, in preparation for validation efforts associated with ICESat-2."*). And based on comments from Reviewer 2, we have added text that specifically quantifies the flat nature of Summit (*"Based on the ATM Level 2 Icessn data product (Krabill, 2010) for all 3 passes associated with the 10 Apr 2014 flight, the slope over the traverse in the along-track direction is 0° and there is a gentle (0.1°) slope in the across-track direction (sloping toward the west); the difference between the maximum and minimum elevations in the vicinity of the traverse, based on the same data product, is only 1.05 m. Given these low slope values, a geolocation error of 10 m is required to achieve a slope-induced elevation error of 0.01 m."*).

To address the limitations of these results, we added a sentence and modified the last paragraph of the Discussion: (*"Results presented here are limited with respect to*

*applicability to the entire ice sheet. As you reach the ice-sheet margins, airborne and satellite laser altimetry data are compromised as a result of, among other environmental variables, surface roughness and slope (Brunt et al., 2010; 2014). However, the ground-based GPS elevation data collected near Summit Station provides a means to characterize airborne elevation data of ATM and LVIS. Comparisons between ATM and LVIS elevations and the ground-based elevations constrain the errors of the airborne datasets. Thus, in situ data, even on short length-scales and over flat surfaces, can form part of a strategy to validate data from airborne, and ultimately satellite, platforms...*")

Below I list a few corrections/suggestions that you will hopefully find useful to revise your paper before it can be published for open discussion and formally peer-reviewed by expert in the field. Note of course that my own assessment during this rapid access review does not anticipate the formal reviews that will follow.

Editor technical corrections/suggestions (Page.Line)

2.2 Here the reader wonders why the bias/precision differ in the two studies? Type of surface? Slope? Elaborate a bit more, maybe in the discussion?
The differences here are associated with 1) terrain and 2) laser campaigns. Language was added to the 'pre-Discuss' draft of this manuscript that addresses this ("*Early assessments of GLAS, based on ground-based Global Positioning System (GPS) surveys of a large and stable salt flat in Bolivia, have shown absolute surface elevation bias of less than 0.02 m and precision of less than 0.03 m under ideal conditions (Fricker et al., 2005). However, estimates of GLAS surface elevation bias and precision from the latter half of the satellite mission have been closer to 0.06 and 0.15 m, respectively (Kohler et al., 2013), based on data from a 10,000 km ground-based GPS traverse in East Antarctica, which included the interior and the margin of the ice sheet.*"). We have subsequently added to this text with some language associated with your previous comment: ("*... which included the interior and the margin of the ice sheet, where surface roughness and slope compromise the accuracy and precision of satellite laser altimetry (Brunt et al., 2010; 2014).*")

7.10 At some point in the method (or result) section the time difference between the field/airborne measurements should be described, briefly. Right now this is first stated in the discussion. This is one of the strength of the study to have such simultaneity so could be emphasized a bit more.
Excellent suggestion. At the end of Section 2.1, we added a paragraph on this ("*When it was logistically possible, the timing of the ground-based survey was coordinated with NASA airborne surveys of the region (Table 1). This allowed for the assessment of airborne lidar performance over ice-sheet interiors (e.g., Brunt et al., 2014). When the timing offset between the airborne and GPS surveys is minimized, assessments of lidar performance are made in the absence of environmental factors (e.g., snow, melt, or wind events) that change the surface and potentially compromise the analysis. From Table 1, six of the airborne campaigns were offset from the ground-based GPS survey*

*by two days or less; however, three of the campaigns were offset by eight days or more, with the maximum offset being 20 days."*).

8.1 can these likely errors be quantified even roughly? In other words, how much of the final error is likely due to the reference data itself.

The error statements here were intended to provide a sense of source. However, you are correct; providing bounds would add to the paper. Quantitatively, our expectation is that this error term is probably very close to the spread, or standard deviation, of the GPS data when comparing it to a repeated ground-based pass (reported in our next response). This value is ~0.06 m, and we do not believe that it can easily be separated into the error components (1 through 5) listed in the text. We have added text that addresses this comment: ("*We compared the second pass to the first pass, using a nearest-neighbor approach, and calculated the mean elevation residual for 1067 points. For the DGPS methods, the TRACK residual was 0.004 m (standard deviation 0.055 m), while the GITAR residual was 0.026 m (standard deviation 0.058 m). For the PPP method, this residual was -0.009 m (standard deviation 0.057 m). Thus, we are confident that the survey methods and data processing techniques associated with the in situ GPS survey provide internally consistent ground-based results. While it is hard to isolate or quantify the non-zero errors associated with the ground-based GPS elevation data, we assume that the 0.055 to 0.059 m range of standard deviations is representative of the contribution of all of the terms mentioned in the previous section.*")

8.17 Could be useful to also provide the standard deviation about the mean for these values.

These values have been parenthetically added to the text (above); they range from 0.055 to 0.058 m.

8.27-31 very impressive statistics! But, as stated in my general comments, this is also very favourable terrain because precisions of the lidar surveys will scaled with surface slope. A limitation that needs to be discussed. Otherwise, others are going to apply your numbers to every OIB flights even over rough topography...

Excellent point. We have added text to the Abstract, Discussion, and Conclusions that points out that these results are on the ice-sheet interior (see response to the general comment).

10.1 "the large surface measurement precision". Do the authors mean high precision or not so good? Ambiguous.

Good point. We changed this phrase to: "*...the relatively poorer quality (0.09 m) surface measurement precision...*"

10.4 "Table 2..". This is a sentence that belongs to the "results"

Good point. We have worked these bias and precision ranges into text within the Results section and removed this sentence from the discussion.

10.16 To what extent a result obtained for a 11 km stripe of data can be extrapolated to 1000s of km? To be discussed

This is a very fair comment; we think it goes with your general comment above. We have added text to the Abstract, Discussion, and Conclusions that points out that these results are on the ice-sheet interior (see response to the general comment). And we have added text that explicitly states why this would break down at the margins (slope/roughness).

11.4 "6000 here "11,000" in the abstract. ???

We have clarified this language. The ground-based component is 11 km; the satellite interests 6 km of the traverse. The language has been changed to: "*
[revised manuscript text omitted]

0.055 ±0.074; $N$ = 255 | 0.005 ±0.073; $N$ = 254
0.005 ±0.074; $N$ = 254 | -0.026 ±0.075; $N$ = 253
-0.026 ±0.075; $N$ = 253 |
| ATM 11 Apr 2012 | 4B/T4 (30°) | 0.067 ±0.045; $N$ = 320
0.067 ±0.045; $N$ = 320 | -0.014 ±0.055; $N$ = 323
-0.014 ±0.055; $N$ = 323 | 0.008 ±0.039; $N$ = 321
0.008 ±0.039; $N$ = 321 |
| ATM 10 Apr 2014 | 4B/T4 (30°) | 0.018 ±0.076; $N$ = 491
0.018 ±0.076; $N$ = 491 | 0.040 ±0.077; $N$ = 491
0.040 ±0.077; $N$ = 491 | -0.021 ±0.075; $N$ = 494
-0.021 ±0.075; $N$ = 494 |
| ATM 02 May 2014 | 4B/T4 (30°) | 0.005 ±0.054; $N$ = 220
0.005 ±0.054; $N$ = 220 | 0.037 ±0.051; $N$ = 221
0.037 ±0.051; $N$ = 221 | -0.005 ±0.052; $N$ = 223
-0.005 ±0.052; $N$ = 223 |
| ATM 09 Apr 2015 [A] | 5A/T3 (30°) | 0.004 ±0.088; $N$ = 470
0.004 ±0.088; $N$ = 470 | -0.026 ±0.087; $N$ = 476
-0.026 ±0.087; $N$ = 476 | -0.064 ±0.087; $N$ = 472
-0.064 ±0.087; $N$ = 472 |
| ATM 09 Apr 2015 [A] | 5B/T5 (5°) | 0.043 ±0.068; $N$ = 365
0.043 ±0.068; $N$ = 365 | 0.015 ±0.070; $N$ = 366
0.015 ±0.070; $N$ = 366 | -0.021 ±0.068; $N$ = 368
-0.021 ±0.068; $N$ = 368 |
| ATM 19 Apr 2016 [B] | 5A/T2 (30°) | -0.070 ±0.075; $N$ = 331
-0.070 ±0.075; $N$ = 331 | -0.043 ±0.073; $N$ = 329
-0.043 ±0.072; $N$ = 329 | -0.108 ±0.059; $N$ = 336
-0.108 ±0.059; $N$ = 336 |
| LVIS 20 Sept 2007 [1] | Pre-OIB | 0.115 ±0.061; $N$ = 1219
0.116 ±0.057; $N$ = 1219 | 0.085 ±0.059; $N$ = 1219
0.086 ±0.056; $N$ = 1219 | 0.081 ±0.061; $N$ = 1218
0.082 ±0.057; $N$ = 1218 |
| LVIS 14 Apr 2010 [1,C] | OIB | 0.037 ±0.064; $N$ = 497
0.035 ±0.060; $N$ = 497 | -0.004 ±0.064; $N$ = 497
-0.006 ±0.060; $N$ = 497 | -0.024 ±0.061; $N$ = 497
-0.027 ±0.058; $N$ = 497 |

Elevation bias (lidar elevation – GPS elevation) and surface measurement precision are in m. DGPS1 is TRACK software processing results, processed to the L1 phase center, 0.056 m above the base plane; DGPS2 is GITAR software processing results, processed to the LC phase center, 0.061 m above the base plane; PPP is Inertial Explorer software processing results, processed to the L1 phase center, 0.056 m above the base plane.

[1] Indicates ITRF00 as the post-processing datum; [2] indicates ITRF05; all others are ITRF08.

[A] Indicates that the instrument was integrated with a C-130; [B] indicates that the instrument was integrated with the NOAA P-3; [C] indicates that the instrument was integrated with the NASA DC-8; all others surveys were flown on the NASA P-3.

kelly brunt 1/3/2017 16:13

kelly brunt 1/5/2017 17:14

kelly brunt 1/6/2017 12:15

kelly brunt 1/6/2017 12:18

kelly brunt 1/3/2017 15:40

kelly brunt 1/5/2017 16:36

kelly brunt 1/6/2017 12:15

kelly brunt 1/6/2017 12:18

kelly brunt 1/3/2017 15:59

kelly brunt 1/5/2017 16:50

kelly brunt 1/6/2017 12:18

kelly brunt 1/13/2017 10:31

kelly brunt 1/6/2017 12:18

[Figure]

[Figure]

kelly brunt 11/15/2016 11:28

**Figure 1:** Map of the Summit Station area, including: representative ground-based GPS survey line (blue line); airborne lidar surveys (gray lines, thicker lines occur earlier); ICESat ground track #0412 (cyan line); ICESat-2 ground tracks (green lines); and Summit Station (red dot). The distance between Summit Station and the southern end of the traverse is ~3 km; the distance between Summit Station and the northern end of the traverse is ~6.5 km. Relative to the trend of ICESat track #0412, the ground-based GPS survey line is oriented both along-track and across-track in order to better characterize the surface slope.

kelly brunt 11/15/2016 11:11
kelly brunt 11/15/2016 11:11

[Figure]

**Figure 2:** The roving GPS antenna, sled, and snowmobile configuration. $GPS_{PC}$ is the surveyed position solution to the phase center of the antenna, $h_{NGSmodel}$ is the NGS model distance between the antenna phase center and the antenna base plane, $h_{AntPost}$ is the height of the antenna post (Table 1), $h_{RunnerDepth}$ is the depth of the sled runners in the snow surface (Table 1), and $h$ is the snow surface (Equation 1).

[Figure]

**Figure 3:** Schematic representation of the lidar measurement strategies and ground-based GPS sample spacing. LVIS measurements (gray): ~10 m diameter footprint and a 1000 m across-track swath width. ATM measurements (green): ~1 m diameter footprint and either a 40 m (narrow-scan; post 2012) or a 250 m (wide-scan) across-track swath width. Ground-based GPS data (blue points) indicate sample spacing. MABEL measurements (black dots) are included to illustrate the limitations of a profiling lidar for this application.

kelly brunt 11/15/2016 16:30

kelly brunt 1/13/2017 10:34

[Figure]

[Figure]

**Figure 4:** Performance of ATM and LVIS over Summit Station through time. Date versus lidar surface bias (m), for ATM and LVIS, for the IceBridge campaigns. Error bars represent surface measurement precision. TRACK DGPS (blue), GITAR DGPS (red), and Inertial Explorer PPP (black) GPS post-processing results are presented. ATM 15 represents the narrow-scan lidar system only. We note that for the GITAR solutions, the base station positions represent an average of four days of data; while for the TRACK solutions, the base station positions represent an average recorded over the duration of the ground-based survey. This may account for the slight offset associated with the TRACK solutions.

kelly brunt 11/15/2016 16:30

kelly brunt 1/4/2017 11:25